# Construction and Evaluation of an Efficient Live Attenuated *Salmonella* Choleraesuis Vaccine and Its Ability as a Vaccine Carrier to Deliver Heterologous Antigens

**DOI:** 10.3390/vaccines12030249

**Published:** 2024-02-27

**Authors:** Xiaoping Bian, Jin Chen, Xin Chen, Chengying Liu, Jianjun Ding, Mengru Li, Xiaofen Zhang, Qing Liu, Qingke Kong

**Affiliations:** 1College of Veterinary Medicine, Southwest University, Tiansheng Road No. 2, Chongqing 400700, China; bxp1212@email.swu.edu.cn (X.B.); cj20211007@email.swu.edu.cn (J.C.); cx0914@email.swu.edu.cn (X.C.); liuda7@email.swu.edu.cn (C.L.); djj150844@email.swu.edu.cn (J.D.); lmr922@email.swu.edu.cn (M.L.); zxf19921218@email.swu.edu.cn (X.Z.); 2Yibin Academy of Southwest University, Southwest University, University Road Section 3, Yibin 644007, China; 3College of Animal Science and Technology, Southwest University, Tiansheng Road No. 2, Yibin 644007, China

**Keywords:** live attenuated *S*. Choleraesuis, monophosphorylated lipid A, heterologous antigens, vaccine carrier

## Abstract

The gram-negative facultative intracellular pathogen *Salmonella enterica* serotype Choleraesuis, also known as *S*. Choleraesuis, is a major financial loss for the pig business. C500 is a vaccine strain that has been used for preventing *S*. Choleraesuis infection in pigs for many years in China. Although it possessed good immunogenicity and protection efficacy, it still showed severe side effects. The truncation of the key gene *rpoS* in C500 was believed to take the major responsibility for its attenuation. To achieve a good balance between attenuation and immunogenicity, *rpoS* was restored to an active state, and other essential virulent genes of *crp*, *fur*, *phoP*, and *aroA* were evaluated for their effects of deletion on safety and immunogenicity. Animal experiments demonstrated that C5001 (C500 *rpoS*^+^ Δ*crp10*) and C5002 (C500 *rpoS*^+^ Δ*fur9*) showed an excellent ability to induce an immune response. To further decrease the endotoxic activity, the combination mutations of Δ*pagL7* Δ*pagP81*::P_lpp_ *lpxE* Δ*lpxR9* were introduced into the mutant strains to generate 1′-dephosphorylated lipid A. Animal experiments showed that SC3 (C500 *rpoS*^+^ Δ*fur9* Δ*pagL7* Δ*pagP81*:: P_lpp_ *lpxE* Δ*lpxR9*) induced higher levels of IgG and secreted IgA antibodies and provided a higher protection rate than SC1 (C500 Δ*pagL7* Δ*pagP81*:: P_lpp_ *lpxE* Δ*lpxR9*) and SC2 (C500 *rpoS*^+^ Δ*crp10* Δ*pagL7* Δ*pagP81*:: P_lpp_
*lpxE* Δ*lpxR9*). We also evaluated the ability of SC3 (C500 r*poS*^+^ Δ*fur9* Δ*pagL7* Δ*pagP81*:: P_lpp_ *lpxE* Δ*lpxR9*) as a vaccine carrier to deliver heterologous protein antigens and polysaccharide antigens. The results indicated that SC3 (C500 r*poS*^+^ Δ*fur9* Δ*pagL7* Δ*pagP81*:: P_lpp_ *lpxE* Δ*lpxR9*) showed an excellent ability to deliver heterologous antigens and induce the host to produce high levels of antibodies. Together, these results indicate that we constructed a safe and efficient attenuated strain of the *S*. Choleraesuis vaccine, which demonstrated strong resistance to infection with wild-type *S*. Choleraesuis and can be employed as a universal vector for the delivery of recombinant antigens.

## 1. Introduction

More than 2000 serotypes of *Salmonella enterica* have been found, many of which are capable of infecting individuals as well as domestic animals with various kinds of diseases. *Salmonella enterica* serovar Typhi (*S*. Typhi) is the pathogen that causes typhoid fever, a systemic sickness that affects humans. Conversely, non-typhoidal *Salmonella* (NTS), such as *Salmonella enterica* serovar Choleraesuis (*S*. Choleraesuis), is one of the primary pathogens causing paratyphoid fever in piglets, often leading to many deaths and huge economic losses for the pig industry due to simultaneous or secondary infection with other pathogens [1]. It is also essential for public health that humans commonly contact *S*. Choleraesuis by eating contaminated poultry, eggs, pork, and beef [2]. The most common clinical symptom of infection in humans is acute gastroenteritis. Young children and older people with low immunity are prone to the further development of bacteremia [3]. Domestic animals have been routinely treated with antibiotics to prevent *S*. Choleraesuis infection. However, the multi-drug-resistant strains of *S*. Choleraesuis are increasing yearly, becoming a severe global problem [4]. Therefore, developing vaccines for humans and animals is an attractive approach for controlling *S*. Choleraesuis infections.

Over the years, several vaccines have been developed specifically for pigs to prevent the infection of *S*. Choleraesuis [5,6,7,8,9]. In China, the C500 vaccine was a live attenuated *S*. Choleraesuis vaccine derived from the C78-3 strain by a chemical approach and licensed for over 50 years to prevent piglet paratyphoid [10]. However, C500 still exhibits a relatively high level of virulence because some pigs experienced severe side-effect reactions after the injected vaccination of C500, showing symptoms of vomiting, shivering, an elevated body temperature, and even death in some cases. Therefore, a safer and more effective vaccine against *S*. Choleraesuis is required. C500 can be continued to attenuate and reduce its virulence, but this process would affect the immunogenicity and alleviate protection efficacy [11]. Hence, continued attenuation of C500 is not a good option for developing the vaccine. We must explore strategies to develop new vaccines that balance safety and immunogenicity.

There are several advantages of live oral attenuated vaccines over other forms of vaccination. Live oral attenuated vaccines induce local immune responses at the mucosal surface and systemic immune responses in the host body. They are easy to manufacture, suitable for large-scale vaccination in pig farms, and do not generate hazardous waste such as needles and syringes [12]. An ideal oral attenuated *Salmonella* vaccine must overcome the unfavorable environment of low nutrients, high acidity, and bile in the stomach and intestine by oral immunization and colonizing the relevant tissues to induce local mucosal humoral immunity in the host [13]. Virulence genes of *Salmonella* have been systematically studied, such as *rpoS*, *crp*, *fur*, *phoPQ*, and *aroA*. RpoS mediates the expression of multiple genes in response to environmental stress (including nutrient deficiency, heat, oxidation, and osmotic shock). It has been reported that RpoS enhances the ability of bacteria to overcome innate host defenses, including stomach acidity and the production of reactive oxygen radicals by immune cells [14,15]. Furthermore, it has been shown that RpoS is essential for the persistence of bacteria in lymphoid organs such as Peyer’s patch (gut-associated lymphoid tissue [GALT]), spleen, and liver, and thus in the initial stages of infection of the host [16,17,18]. This led us to hypothesize that the presence of functional RpoS in attenuated *S*. Choleraesuis would enhance the viability of the strains in mice because of their ability to overcome the gastrointestinal environment in the host and enter the mucosa-associated lymphoid tissue. The *crp* gene encodes the cAMP receptor protein, which plays a crucial role in regulating glucose and amino acid metabolism, as well as controlling the production of pili and flagella [19]. It has been reported that a *Salmonella* mutant missing the *crp* gene had reduced virulence in mice and demonstrated effective protection against wild-type *Salmonella* infection following immunization [20]. Fur is a global regulator, which plays an essential role in iron uptake and acid resistance [21,22,23], and *S*. Typhimurium, with the deletion of the *fur* gene, can confer cross-protection against multiple *Salmonella* in mice [24]. PhoP is a transcriptional activator that controls the expression of genes responsible for virulence and intracellular survival of *Salmonella* within macrophages [25]. The *phoP* mutant is capable of eliciting a protective immune response in mice [26,27]. The *aroA* gene is involved in the production of aromatic amino acids, which are rare in the host. Consequently, the removal of the *aroA* gene leads to a *Salmonella* phenotype characterized by a lack of essential nutrients [28]. Attenuated *Salmonella* carrying the *aroA* mutant also protected mice against a fatal dose of infection of wild-type *Salmonella* [29].

Lipopolysaccharides (LPS) are present on the outer membranes of nearly all gram-negative bacteria and consist of lipid A, core oligosaccharide, and O-antigen polysaccharide [30,31]. Lipid A is essential for activating TLR4-related pathways and Caspase 4/5/11-dependent pathways [32]. With long-term evolution, *Salmonella* improves its ability to survive in different environments by modifying the lipid acyl chains and decorating phosphate groups. Previous studies indicated that 3-O-deacylated 4′-monophosphoryl lipid A (3D-MPL) can still induce dendritic cell maturation and Th1-biased immune responses while showing low endotoxicity as a vaccine adjuvant [33]. Our previous study showed that LpxE selectively removes the 1-phosphate group of lipid A in *Salmonella*, resulting in a product that closely resembles monophosphoryl lipid A (MPL) [31]. In addition, we demonstrated that the presence of the *lpxE* gene on the chromosome did not lead to a reduction in the ability of attenuated *Salmonella* with additional gene deletions to deliver heterologous antigens [31]. Therefore, to improve the safety and maintain immunogenicity, we introduced the *lpxE* gene into the genome of attenuated *S.* Choleraesuis.

In the present study, we developed a novel low endotoxin activity recombinant attenuated vaccine SC3 (C500 *rpoS*^+^ Δ*fur9* Δ*pagL7* Δ*pagP81*:: P_lpp_
*lpxE* Δ*lpxR9*) based on the vaccine strain C500 of *S*. Choleraesuis. We evaluated its attributes in vitro and in vivo by quantifiably comparing them to C500 to show attenuation and safety. We also assessed the ability of SC3 (C500 *rpoS*^+^ Δ*fur9* Δ*pagL7* Δ*pagP81*:: P_lpp_
*lpxE* Δ*lpxR9*) as a vector to deliver heterologous antigens (GDH from *S. suis* and O-antigen polysaccharide of *E. coli* O9) for inducing immune responses in mice.

## 2. Material and Methods

### 2.1. Growth Conditions, Medium, Plasmids, and Strains of Bacteria

Table 1 provides a comprehensive list of all the bacterial strains and plasmids used in this investigation. *S*. Choleraesuis was routinely cultured either in Luria-Bertani (LB) broth (L1010, Solarbio Biological Reagent Co., Ltd., Beijing, China) or on LB agar (L1015, Solarbio Biological Reagent Co., Ltd., Beijing, China) at a temperature of 37 °C [34]. In some cases, if necessary, the media were supplemented with the following concentrations of antibiotic: 50 μg/mL for kanamycin (A600286, Sangon Biological Reagent Co., Ltd., Shanghai, China) and 25 μg/mL for chloramphenicol (A600118, Sangon Biological Reagent Co., Ltd., Shanghai, China). To facilitate the growth of Asd- strains, DAP was included at a concentration of 50 µg/mL (D1377, Sigma-Aldrich Chemical Reagent Co., Ltd., WI, USA). For the cultivation of strains involving allelic exchange experiments, *sacB* gene-based counterselection was conducted on LB agar containing 5% sucrose (A610498, Sangon Chemical Reagent Co., Ltd., Shanghai, China). Additionally, N-minimal medium [35] supplemented with 0.1% casamino acids (113060012, MP Biomedicals Biological Reagent Co., Ltd., CA, USA), 38 mM glycerol (A100854, Sangon Biological Reagent Co., Ltd., Shanghai, China), and 10 mM MgCl_2_ (10012818, Sinopharm Chemical Reagent Co., Ltd., Shanghai, China) was utilized.

### 2.2. Construction of Plasmids and Mutant Strain

The allelic exchange method was utilized to introduce gene mutations in *S*. Choleraesuis, employing pYA4278 as described in the previous study [36]. The transformation of *S*. Choleraesuis or *E. coli* was carried out using a technique known as electroporation. To select the transformants, LB agar plates supplemented with suitable antibiotics were employed. Specifically, the selection for Asd+ plasmids was performed on LB agar plates. A comprehensive list of the primers employed in this investigation can be found in Appendix A. To restore the *rpoS* gene, the C3545 genome was used as the template for cloning. We employed two sets of primers, namely, *rpoS*-F/*rpoS*-R, to amplify a DNA fragment of approximately 1656 bp containing the entire *rpoS* gene open-reading frame (ORF), the region upstream and the region downstream of the *rpoS* gene from the C3545 genome. The PCR products were purified by agarose gel (A620014, Sangon Biotech, China) and ligated to pYA4278 using Gibson Assembly Master Mix (E55510, New England Biolabs Biological Reagent Co., Ltd., MA, USA) to generate the suicide plasmid pSW001. Then the *rpoS* gene was introduced into C500 via allelic exchange by conjugation with the *E. coli* strain χ7213 [40] harboring the suicide plasmid of pSW001, to generate the strain C5000 (C500 *rpoS*^+^). Subsequently, the genes of *crp*, *fur*, *phop*, and *aroA* were introduced into C5000 (C500 *rpoS*^+^), respectively. In short, to delete the *crp* gene from the C5000 (C500 *rpoS*^+^) genome, a 253 bp upstream DNA region and a 253 bp downstream DNA region were precisely amplified using PrimeSTAR Max DNA Polymerase (R045A, Takara Biotech Biological Reagent Co., Ltd., Kyoto, Japan) from the C500 genome by applying primers *crp*-1F/*crp*-1R and *crp*-2F/*crp*-2R, respectively. These two DNA segments were then fused via primer *crp*-1F and *crp*-2R with overlap PCR. The fused PCR products were purified by agarose gel (A620014, Sangon Biotech Biological Reagent Co., Ltd., Shanghai, China) and cloned into pYA4278 utilizing Gibson Assembly Master Mix to construct a new suicide plasmid (pSW002). The *crp* gene was introduced into C5000 (C500 *rpoS*^+^) via allelic exchange by conjugation with the *E. coli* strain χ7213 [40] containing the suicide plasmid of pSW002, resulting in the *crp*-deficient strain C5001 (C500 *rpoS*^+^ Δ*crp10*). The same approach was used to delete the *fur*, *phop*, and *aroA* genes to generate the plasmids pSW003, pSW004, and pW005 and then generate the corresponding deficient strains C5002 (C500 *rpoS*^+^ Δ*fur9*), C5003 (C500 *rpoS*^+^ Δ*phoP11*), and C5004 (C500 *rpoS*^+^ Δ*aroA12*), respectively.

### 2.3. Phenotypic Determination of Bacteria

Bacterial strain phenotypes were identified in vitro, and each experiment was conducted not less than twice. The growth of *Salmonella* mutants was assessed in LB broth or LB broth containing 6% NaCl. Briefly, a single bacterial strain clone was selected, placed into LB broth, and cultured for the entire night at 37 °C and 180 rpm. The following day, the overnight culture was incubated under the same conditions after being diluted 100 times into the corresponding fresh medium. The bacterial solution’s optical density value at 600 nm (OD_600_) was determined every hour. Finally, the recorded results were summarized, and the growth curves of all mutant strains were plotted via the Graph-Pad Prism 8.0 software. As previously described, outer membrane proteins (OMPs) [38] and LPS [42,43] were isolated from *S*. Choleraesuis. The OMPs underwent electrophoresis on a sodium dodecyl sulfate-polyacrylamide gel (SDS-PAGE) and were subsequently treated with Coomassie brilliant blue. The bacterial strains of *S*. Choleraesuis went through silver staining to establish their LPS profile following the method outlined by Hitchcock and Brown [44]. The swimming motility of the bacteria was evaluated by employing LB plates that were solidified using 0.3% agar (wt/vol), as previously delineated [45]. In short, 0.3% agar LB plates were inoculated with 5 μL of bacterial suspension (roughly 8 × 10^6^ CFU/mL) and then incubated for 8 h at 37 °C in an incubator. A ruler was used to determine the swimming diameter.

### 2.4. Animals

Female BALB/c mice (18–19 g) and New Zealand white rabbits were obtained from Chengdu Dashuo Laboratory Animal Technology Co., Ltd., Chengdu, China. To protect the welfare of the animals, care was given to them, and the tests were carried out in compliance with the recommendations made in the “Guide for the Care and Use of Laboratory Animals”. The animal house’s temperature, humidity, and ventilation were regularly checked to guarantee ideal climatic conditions. The mice received appropriate care and were managed by competent and trained personnel. Every attempt was carried out to mitigate animal suffering while the course of the experiments was being conducted. Following their arrival, a seven-day acclimation period was provided for all animals prior to the commencement of experiments.

### 2.5. Daily Observation of Mice

Daily observations were made on mice that received vaccinations. Monitoring was conducted for local reactions and any adverse effects, including abnormalities, a decline in overall health, and a decrease in food consumption. Additionally, evidence of unkempt fur, restlessness, diarrhea, morbidity, and mortality was recorded. 

### 2.6. Immunity of Mice

Mice were immunized by oral administration. Briefly, 5 mL of appropriate fresh media was used to stationary culture a solitary bacterial strain clone overnight in a 37 °C incubator. On the following day, the cultures were diluted 1:100 within a 100 mL identical medium and then incubated at 37 °C until the OD_600_ reached 0.85 (approximately 1 × 10^9^ CFU/mL). The 100 mL suspension of bacterial strain was gathered through centrifugation at 5000 rpm under ambient temperature conditions and resuspended in 2 mL of buffered saline with gelatin (BSG), at which point the bacterial concentration was approximately 1 × 10^9^ CFU/20 µL. The mouse cage was removed from food and water for a duration of 6 h, and the animals were orally given 20 µL of BSG, including 10^9^ CFU of the corresponding mutants. At 5 weeks, mice received a subsequent dose of the identical strain, matching the previous dosage.

### 2.7. Localization of S. *Choleraesuis* in Organs

To assess the bacterial load in organs after the mice were orally given 20 µL of BSG containing 1 × 10^9^ CFU mutant strains, at specific time intervals, three mice from each group were humanely euthanized. The researchers gathered Peyer’s patches (PP) from various locations on the surface of the small intestine. The bacterial load within the PP indicated the average bacterial load in the combined PP from every mouse. Samples from the spleen and liver were obtained and weighed individually. To obtain a homogenized sample, each sample was added to 1 mL of phosphate buffer saline (PBS). Subsequently, various dilutions of the samples ranging from 10^−1^ to 10^−3^ (depending on the specific tissue) were applied onto MacConkey agar and LB agar to measure the total amount of live bacteria present. MacConkey agar indicator plates are designed to exclude interference from *E. coli* in tissue homogenates. The bacterial loading was calculated based on the bacteria on the indicator plates. If there were no colonies observed on the LB plate after culturing the homogenizing 0.1 mL of the organ, the selenite cysteine broth (100212, Sigma-Aldrich Biological Reagent Co., Ltd., WI, USA) was used to inoculate the remaining PBS solution from each tissue sample the following day. Samples that showed positive results after being enriched in selenite cysteine broth at 37 °C overnight were noted to have a CFU count of less than 10 per gram.

### 2.8. Determination of Virulence of Mutants in Mice

To measure the 50% lethal dose (LD_50_), C500 or C5000 (C500 *rpoS*^+^) strains were grown to an OD_600_ of about 0.85 (~10^9^ CFU/mL) in 100 mL of LB broth at 37 °C. After centrifuging the strains at 5000 rpm under room temperature, the strains were suspended in 2 mL of BSG (~10^9^ CFU/20 µL) and then adjusted at densities suitable for immunization. Mice were randomly divided into eight groups and orally infected with C500 or C5000 (C500 *rpoS*^+^) at doses of 7 × 10^9^ CFU/20 µL, 3.5 × 10^9^ CFU/20 µL, 1.75 × 10^9^ CFU/20 µL, and 7 × 10^8^ CFU/20 µL, respectively. Every eight were infected with the same dose. Before being inoculated, all mice were fasted for 6 h. Mice were monitored daily for evidence of unkempt fur, restlessness, diarrhea, and morbidity, and mortality was recorded for 25 days after the challenge. The lethal dose 50 was computed using probit analysis on the SPSS 26.0 software [46].

### 2.9. Tissue Damage Caused by S. Choleraesuis

To investigate the spleen tissue damage caused by the mutants after the mice were orally given 20 µL of BSG containing 1 × 10^9^ CFU mutant strains, we employed Hematoxylin and eosin (H&E) staining for histopathological analysis. On the sixth day following oral inoculation, two mice in each group were humanely euthanized and the spleen was taken to determine the scope of the organ damage brought by the mutants. The standard procedure was followed for fixing and processing tissues for H&E staining. We observed the morphological aspects of tissues to identify signs of inflammation, including infiltration of macrophages, accumulation, distortion, and abnormal red pulp areas in the spleen.

### 2.10. Ileal Loop Experiment on Rabbits

After an overnight fast, New Zealand white rabbits were given Zoletil to anesthetize through an ear vein. Ligate the ileum at 1 cm intervals to create loops 3–5 cm long. Separate loops were injected with mutant strains at a titer of 1 × 10^9^ CFU, with a 1 mL volume. As a control, one of the loops was injected with LB broth. Closure of the abdominal musculature involved the use of 3-0 chromic gut sutures, followed by closure of the skin using 3-0 Ethilon sutures. Throughout the experiment, the rabbits were maintained at 37 °C using a thermal blanket. After 8 h, the rabbits were killed by intravenous air injection. The intestinal segment was ligated by laparotomy, and the intestinal tissue was fixed with 10% formalin for pathological section analysis by H&E staining.

### 2.11. Survival Assay

The protection rates of the attenuated *S*. Choleraesuis were assessed at 9 weeks post-immunization by the oral challenge of mice with 10^9^ (~10,000 × LD_50_) wild-type *S*. Choleraesuis C3545 in 20 μL BSG. C3545 is a clinical isolate from a pig with an LD_50_ of 10^5^ CFU/20 μL [41,47]. Mice were monitored daily for evidence of unkempt fur, restlessness, diarrhea, and morbidity, and mortality was recorded for 25 days after the challenge.

### 2.12. Quantitative Analysis of Specific Antibody Levels by ELISA

Serum was collected by puncturing the mandibular vein to obtain blood, followed by centrifugation at 3500 rpm. To analyze the secretory IgA (S-IgA), the vaginal tract of each mouse was washed with 60 μL PBS, and the wash fluids were pooled together. The enzyme-linked immunosorbent assay (ELISA) was utilized to quantify the concentration of serum and vaginal wash antibodies against *Salmonella* LPS and OMPs induced by attenuated *Salmonella* strains, following established protocols [44,45]. Briefly, solutions containing 200 ng per well of *S*. Choleraesuis-derived OMP or LPS were suspended in 100 µL of coating buffer composed of sodium carbonate-bicarbonate (pH 9.6). These solutions were then utilized to coat 96-well plates for an extended duration of time at a temperature of 4 °C. To generate standard curves for each antibody isotype, we coated plates with duplicate samples of purified mouse Ig isotype standard (1010-01, Southern Biotech Biological Reagent Co., Ltd., AL, USA). Each well was coated with 200 ng of the standard in 100 µL of coating buffer. The plates were first washed three times with tris buffered saline with 0.1% Tween 20 (TBST) and then blocked with a solution of 3% BSA (A602449, Sangon Biotech Biological Reagent Co., Ltd., AL, USA) for 2 h at 37 °C. Next, a 100 µL volume of the sample, diluted 100-fold or 10-fold, was added to each well in duplicate. The plates were then incubated for 1 h at 37 °C. To the standard curve, 100 µL of unconjugated IgG (0107-01, Southern Biotech Biological Reagent Co., Ltd., AL, USA), IgA (0106-01, Southern Biotech Biological Reagent Co., Ltd., AL, USA), IgG1 (0102-01, Southern Biotech Biological Reagent Co., Ltd., AL, USA), or IgG2a (0103-01, Southern Biotech Biological Reagent Co., Ltd., AL, USA) obtained from ordinary mice were diluted in a 2x serial in PBS. The unconjugated IgG was diluted from 500 ng/mL to 0.488 ng/mL, IgG1, and IgG2a from 1 µg/mL to 8 ng/mL, and the IgA was diluted from 500 ng/mL to 0.488 ng/mL. Following TBST washing, each well was added with a 1:5000 dilution of biotinylated goat anti-mouse IgA (1040-08, Southern Biotech Biological Reagent Co., Ltd., AL, USA), IgG1 (1070-08, Southern Biotech Biological Reagent Co., Ltd., AL, USA), IgG2a (1080-08, Southern Biotech Biological Reagent Co., Ltd., AL, USA), or IgG (1030-08, Southern Biotech Biological Reagent Co., Ltd., AL, USA). The plates were then incubated at 37 °C for 1 h. A 100 µL volume of p-nitrophenyl phosphate (N1891, Sigma-Aldrich Biological Reagent Co., Ltd., WI, USA) with a final concentration of 1 mg/mL was added to each of the wells for color development after they were previously incubated for an hour at 37 °C with a 1:3000 dilution of streptavidin-alkaline phosphatase conjugate (7100-04, Southern Biotech Biological Reagent Co., Ltd., AL, USA). Color development (absorbance) was read at 405 nm using an automated ELISA plate after appropriate incubation. Using linear regression in Excel (R^2^ ≥ 0.95), the OD values at 405 nm were plotted against the representative concentrations of the diluted unconjugated antibody solutions to generate the standard curves. The corresponding standard curve was applied to calculate the total levels of antibodies in the samples.

### 2.13. Analysis of pSW-GDH and pSW-O9 Expression by Western Blot

To evaluate the heterologous antigen synthesis, the recombinant attenuated *S*. Choleraesuis strain SC3 Δ*asd* harboring the GDH protein and O-antigen of *E. coli* O9 were cultured overnight in LB medium with aeration at 37 °C. One milliliter of culture was collected from each passage and prepared for Western blot analysis. Western blot analysis was performed as previously described using anti-GDH and anti-*E. coli* O9 antisera [37,39]. The following antibody was goat anti-rabbit immunoglobulin G (SAB4600068, Sigma-Aldrich Biological Reagent Co., Ltd., WI, USA), which was conjugated with horseradish peroxidase.

### 2.14. Statistical Analysis

The Graph-Pad Prism 8.0 software was utilized to conduct statistical calculations. Unless otherwise mentioned, numerical data were presented as means ± SEM. To determine the varying significance of bacterial motility, one-way or two-way ANOVA analysis was performed, followed by the application of Tukey’s multiple comparisons test. The log-rank test was implemented to assess differences in mouse survival, with the Kaplan–Meier survival curve serving as the monitoring tool. To compare means, the least significant difference test was employed. A *p*-value of less than 0.05 was regarded as indicative of a significant difference.

## 3. Results

### 3.1. Construction and Biological Characteristics of the C5000 Strain

To develop a new vaccine that maintains superior immunogenicity with moderate virulence based on the C500 strain, the complete *rpoS* gene was introduced into the corresponding position of the C500 genome by the homologous recombination method (Figure 1). The *rpoS* gene insertion in C500 was verified by PCR using the primers *rpoS*-F and *rpoS-*R with the expected amplicon size of 1656 bp from the wild-type C3545 and was confirmed by sequencing; the resultant mutant strain was named C5000 (C500 *rpoS*^+^).

As a gastrointestinal pathogen, *Salmonella* has developed a variety of survival strategies to withstand harsh environments. After oral consumption, *Salmonella* upregulates many amino acid decarboxylase systems and synthesizes acid-activated proteins, including RpoS [18] and PhoPQ [27], to avoid an acidic pH and maintain pH homeostasis in the stomach. Therefore, we detected the growth curve of C5000 (C500 *rpoS^+^*) in the hypertonic LB broth (LB containing 6% NaCl). The results showed that C5000 (C500 *rpoS^+^*) grew faster than C500 (Figure 2A). In addition, we also monitored the growth curve of C5000 in N-minimal medium and N-minimal medium with different pH. The results indicated that the growth was consistent with that in the 6% NaCl LB broth (Appendix A).

### 3.2. Virulence Analysis of the C5000

To evaluate the effect of the *rpoS* gene in an activated state on the C500 genome in mice, this study compared the colonization and invasive capacity of C500 and C5000 (C500 *rpoS^+^*) in the PP, spleen, and liver tissues of infected mice, as well as the LD_50_ of C5000 (C500 *rpoS^+^*). The results showed that the colonization capacity of C5000 (C500 *rpoS^+^*) in the PP tissues was significantly higher than C500 strains at 3 days after inoculation and exhibited a similar level at 7 and 18 days. Still, the colonization ability of C5000 (C500 *rpoS^+^*) in the liver and spleen was significantly higher than C500 during the tested periods (Figure 2B). The histopathological analysis showed that a small number of neutrophils were present in the splenic tissue of the C500 group. In contrast, apparent neutrophil infiltration and forming infiltrative foci were observed in the C5000 (C500 *rpoS^+^*) group (Figure 2C). The virulence of C5000 (C500 *rpoS^+^*) in the BALB/c mice was only slightly higher than that of C500 (Appendix A). These results indicated that the *rpoS* gene was in an activated state in the C5000 (C500 *rpoS^+^*), contributing to enhanced colonization in mice.

### 3.3. Phenotype Determination and Histopathological Analysis Induced by C5001, C5002, C5003, and C5004

The essential virulent genes of *crp*, *fur*, *phoP*, and *aroA* were deleted from C5000 (C500 *rpoS*^+^) to yield C5001 (C500 *rpoS*^+^ Δ*crp10*), C5002 (C500 *rpoS*^+^ Δ*fur9*), C5003 (C500 *rpoS*^+^ Δ*phoP11*), and C5004 (C500 *rpoS*^+^ Δ*aroA12*). We evaluated the phenotypes of each mutant strain growing in LB broth or agar plate at 37 °C (Figure 3A). All the strains showed similar growth rates except C5001 (C500 *rpoS*^+^ Δ*crp10*), which grew slowest, and no differences were observed for each mutant strain derived from C500, but they were significantly slower than the wild-type strain C3545 in the motility test (Figure 3B). The SDS-PAGE gel indicated that the OMP profiles exhibited similar bands among all the strains, including C500 (Figure 3C). There were no apparent differences among the LPS profiles of these strains detected by silver staining (Figure 3D). 

To understand whether the *crp*, *fur*, *phoP,* and *aroA* mutations have any effect on inducing inflammation in the BALB/c mice, histopathological analysis was performed after the mutants infected mice 6 days later. The spleen of mice inoculated by C500, C5001 (C500 *rpoS*^+^ Δ*crp10*), C5002 (C500 *rpoS*^+^ Δ*fur9*), and C5003 (C500 *rpoS*^+^ Δ*phoP11*) showed some inflammatory foci but did not show severe pathological changes in the infected mice (Figure 4A). The results indicated that these *S*. Choleraesuis mutant strains exhibited low virulence during the experiment time. However, the spleen of mice infected by C5004 (C500 *rpoS*^+^ Δ*aroA12*) had many neutrophils infiltrated. The infected mice showed ruffled fur, depression, weight loss, and even death, indicating that C5004 (C500 *rpoS*^+^ Δ*aroA12*) is not attenuated sufficiently enough to be a vaccine candidate. Therefore, we excluded C5004 (C500 *rpoS*^+^ Δ*aroA12*) in the subsequent study. As lipid A is its key stimulator to induce an inflammatory response in the host; thus, some mutations related to lipid A modification were introduced to reduce its endotoxic activity.

### 3.4. Immunogenicity Assessment Induced by C5001, C5002, C5003, and C5004

To test the immunogenicity of the *S*. Choleraesuis mutant strains, the antibody responses against OMPs and LPS from *S*. Choleraesuis in mice were measured. Mice were immunized orally (10^9^ CFU/20 µL) at 5 weeks after the initial immunization. Blood and vaginal secretions were collected at 8 weeks after the initial immunization to detect S-IgA and IgG antibody levels. The results showed that higher serum S-IgA and IgG antibody levels against OMPs from *S*. Choleraesuis were detected in the mice immunized with the *S*. Choleraesuis mutant strains compared to those with BSG (Figure 4B,C). The serum IgA and IgG levels against OMPs from *S*. Choleraesuis in mice immunized with C5003 (C500 *rpoS*^+^ Δ*phoP11*) were similar to those in mice immunized with C500. Notably, the serum antibody levels of S-IgA and IgG against OMPs and LPS from *S*. Choleraesuis in mice immunized with C5001 (C500 *rpoS*^+^ Δ*crp10*) and C5002 (C500 *rpoS*^+^ Δ*fur9*) were significantly higher than those in mice immunized with C500 and C5003 (C500 *rpoS*^+^ Δ*phoP11*). These results indicated that all *S*. Choleraesuis mutant strains could induce high levels of S-IgA and IgG antibodies against OMPs and LPS from *S*. Choleraesuis in mice, especially C5001 (C500 *rpoS*^+^ Δ*crp10*) and C5002 (C500 *rpoS*^+^ Δ*fur9*).

To determine whether the *S*. Choleraesuis mutant strains protect mice against *S*. Choleraesuis infection, mice were orally challenged with 20 µL of a lethal dose of wild-type *S*. Choleraesuis C3545 containing 10^9^ (10000 × LD_50_) at 9 weeks after initial immunization. The results showed that all negative control mice died during the observation time after the C3545 challenge, while the C500, C5001 (C500 *rpoS*^+^ Δ*crp10*), and C5002 (C500 *rpoS*^+^ Δ*fur9*) strains conferred 70%, 50%, and 60% protection rates, respectively (Figure 4D). 

### 3.5. Construction of S. *Choleraesuis* Mutant Strains with Low Endotoxic Activity and Evaluations of Immunogenicity and Protection Rate

Our previous study showed that LpxE selectively removes the 1-phosphate group of lipid A in *Salmonella*, generating a product similar to monophosphoryl lipid A (MPL) [31]. In this study, to further decrease the endotoxic activity, the combination mutations, including Δ*pagL7* Δ*pagP81*::P_lpp_ *lpxE* Δ*lpxR9*, which will generate 4′-monophosphorylated lipopolysaccharide, were introduced into the mutants of *S*. Choleraesuis, resulting in SC1 (C500 Δ*pagL7* Δ*pagP81*::P_lpp_ *lpxE* Δ*lpxR9*), SC2 (C500 *rpoS*^+^ Δ*crp10* Δ*pagL7* Δ*pagP81*::P_lpp_
*lpxE* Δ*lpxR9*), and SC3 (C500 *rpoS*^+^ Δ*fur9* Δ*pagL7* Δ*pagP81*::P_lpp_ *lpxE* Δ*lpxR9*), respectively. These mutations did not impact their growth in LB media (Appendix A).

We also assessed the induced inflammation in mice after the mutants harboring Δ*pagP81*::P_lpp_ *lpxE* Δ*lpxR9* infected mice 7 days later; no noticeable histopathological changes were observed (Appendix A). To further evaluate the endotoxic activity of these mutant strains, the rabbit ileal loops were used for this purpose. After being injected into ligated loops, mutant strains were cultured for eight hours. H&E-stained histological samples from loops treated with several mutant strains were viewed under a microscope. The C5002 (C500 *rpoS*^+^ Δ*fur9*) strain caused severe damage to the mucosa of mice, resulting in a large necrotic infiltration of the epithelium and a large infiltration of polymorphonuclear leukocytes (PMN). In contrast, the mutant strain SC3 (C500 *rpoS*^+^ Δ*fur9* Δ*pagL7* Δ*pagP81*::P_lpp_
*lpxE* Δ*lpxR9*) induced some tissue destruction and no apparent PMN infiltration (Figure 5), indicating that the 1′-dephosphorylated lipid A in *S*. Choleraesuis aids in a reduction in the inflammatory response.

To evaluate the immune response induced by *S*. Choleraesuis mutant strains in mice, serum and vaginal secretions were collected to detect S-IgA and IgG antibody levels 8 weeks after the primary immunization. As shown in the results (Figure 6A–D), antibody levels of mice immunized with the SC3 (C500 *rpoS*^+^ Δ*fur9* Δ*pagL7* Δ*pagP81*::P_lpp_
*lpxE* Δ*lpxR9*) were significantly higher than those in the control group and other groups. The results indicated that *S*. Choleraesuis mutant strains with 1′-dephosphorylated lipid A could reduce bacterial endotoxin while maintaining the induced immune response in mice.

To determine whether the *S*. Choleraesuis mutant strains with low toxicity protect mice against *S*. Choleraesuis infection, mice were orally challenged with 20 µL of a lethal dose of wild-type *S*. Choleraesuis C3545 containing 10^9^ (10,000 × LD_50_) at 9 weeks after initial immunization. The results showed that SC3 (C500 *rpoS*^+^ Δ*fur9* Δ*pagL7* Δ*pagP81*::P_lpp_
*lpxE* Δ*lpxR9*) conferred 80% protection, and C5002 (C500 *rpoS*^+^ Δ*fur9*) provided 60% protection (Figure 6E).

### 3.6. Evaluations of SC3 as a Vaccine Carrier to Deliver Heterologous Antigens

To evaluate the ability of SC3 (C500 *rpoS*^+^ Δ*fur9* Δ*pagL7* Δ*pagP81*::P_lpp_
*lpxE* Δ*lpxR9*) as a vector to deliver heterologous antigens, we selected the GDH protein antigen of *S. suis* and the O-polysaccharide antigen of *E*. *coli* O9. GDH is a glutamate dehydrogenase protein on the surface of *S*. *suis*, an important virulence factor and effective antigen of *S. suis*. The O-polysaccharide is a potent protective antigen of gram-negative bacteria. In this study, we evaluated the ability of SC3 (C500 *rpoS*^+^ Δ*fur9* Δ*pagL7* Δ*pagP81*::P_lpp_
*lpxE* Δ*lpxR9*) as a vector to deliver the GDH protein antigen and O-polysaccharide antigen. The results showed that the 48 kDa GDH protein and O-antigen of *E. coli* O9 were detected in SC3 Δ*asd* (C500 *rpoS*^+^ Δ*fur9* Δ*pagL7* Δ*pagP81*::P_lpp_
*lpxE* Δ*lpxR9* Δ*asd*) (Figure 7A–C), indicating that the plasmids in SC3 Δ*asd* (C500 *rpoS*^+^ Δ*fur9* Δ*pagL7* Δ*pagP81*::P_lpp_
*lpxE* Δ*lpxR9* Δ*asd*) are stable and able to synthesize both protein antigens and O-polysaccharide antigens.

To evaluate its efficacy in delivering heterologous antigens by SC3 (C500 *rpoS*^+^ Δ*fur9* Δ*pagL7* Δ*pagP81*::P_lpp_
*lpxE* Δ*lpxR9*), SC3 Δ*asd* (C500 *rpoS*^+^ Δ*fur9* Δ*pagL7* Δ*pagP81*::P_lpp_
*lpxE* Δ*lpxR9* Δ*asd*) carrying pSW-GDH and pSW-O9 were used to vaccinate the mice by oral administration. The results showed that all the administered mice produced high antibodies, including anti-GDH and O-antigen levels of *E*. *coli* O9 (Figure 7D). The results indicated that SC3 Δ*asd* (C500 *rpoS*^+^ Δ*fur9* Δ*pagL7* Δ*pagP81*::P_lpp_
*lpxE* Δ*lpxR9* Δ*asd*) could be used as a vaccine vector to deliver heterologous protein antigens and polysaccharide antigens and is able to induce immune responses against exogenous antigens.

## 4. Discussion

Salmonellosis caused by *S*. Choleraesuis in pigs leads to substantial economic losses in a large-scale pig farm. Clinical symptoms of this disease often show fever, depression, septicemia, arthritis, diarrhea, and even death in some cases [1]. As these bacteria often spread quickly in industrialized pig farms by contaminated food and water, the vaccine is an efficient approach to prevent epidemics and control salmonellosis. C500 is a vaccine strain that has been widely used for preventing *S*. Choleraesuis infection in pigs for many years in China [48,49]. Although C500 effectively prevents salmonellosis caused by *S*. Choleraesuis, it still shows a high level of side effects. The previous results indicated that further attenuation of C500 is not a good option for developing a live attenuated vaccine because this process would reduce its immunogenicity and protection efficacy [11]. Therefore, we need to explore strategies to develop new vaccines with a good balance between safety and immunogenicity. In this work, we re-attenuated C500 to construct a safer and more immunogenic attenuated vector of *S*. Choleraesuis.

The previous genome sequence results showed that many genes were lost or inactivated in C500, including *asr*, *ydgF*, *ydgD*, *ydgE*, *rpoS*, and *ptsG* [50]. The animal experiments indicated that the inactivation of the *rpoS* gene, a vital transcriptional regulator playing an essential role in *Salmonella* infection, was the main reason for its attenuation of C500 [10]. Meanwhile, studies with *S*. Typhimurium and the mouse model of typhoid fever also demonstrated that RpoS played an essential role in mice colonization and survival [16,17,18]. Considering these factors, the strategies to generate an effective vector of *S*. Choleraesuis were (1) to restore functional *rpoS* genes for enhancing the viability of the strains in mice because of their ability to overcome the gastrointestinal environment in the host and enter the mucosa-associated lymphoid tissue and (2) to remove essential virulence genes and to construct *S*. Choleraesuis vectors capable of delivering heterologous antigens. As we expected, C5000 (C500 *rpoS*^+^) was more adapted to the gastrointestinal environment, thus contributing to colonization in mice (Figure 2B), although C5000 (C500 *rpoS*^+^) did not exhibit significant virulence enhancement compared with C500 in mice (Appendix A). This result indicates that functional RpoS in C5000 (C500 *rpoS*^+^) can increase the persistence of bacteria in lymphoid tissues, which is consistent with previous studies [16,17,18]. Despite earlier reports showing that the loss of functional RpoS was the main reason for the attenuation of C500 [10], we found that C5000 (C500 *rpoS*^+^) did not restore to a wild-type virulence state in our detecting virulence in mice. This may be due to the inactivation of multiple other genes in addition to RpoS in C500, which collectively affect the virulence of C500.

We aimed to construct an attenuated *S*. Choleraesuis vaccine strain that has a persistent ability to stimulate intense primary and maintain lasting memory immune responses while retaining fewer side effects. In this work, we analyzed the general characteristics of *S*. Choleraesuis mutation strains both in vitro and in vivo. In the in vitro studies, the phenotypes of the *S*. Choleraesuis mutant strains were confirmed and were precisely the same as those in *S*. Typhimurium strains as we expected (Figure 3). In terms of safety based on the pathological analysis of the spleen, severe hyperemia and a massive inflammatory exudation in mice immunized with C5004 (C500 *rpoS*^+^ Δ*aroA12*) were observed; moderate hyperemia and a small amount of inflammatory exudation in mice immunized with C500 and C5002 (C500 *rpoS*^+^ D*fur9*) and mild histopathological lesions in mice immunized with C5001 (C500 *rpoS*^+^ D*crp10*) and C5003 (C500 *rpoS*^+^ D*phoP11*) (Figure 4A) were also observed. Notably, some mice died after C5004 (C500 *rpoS*^+^ Δ*aroA12*) was administered to them, indicating that C5004 (C500 *rpoS*^+^ Δ*aroA12*) is not attenuated sufficiently enough to be a vaccine candidate. Therefore, we excluded C5004 (C500 *rpoS*^+^ Δ*aroA12*) in the subsequent study. Although colonization can reflect the interaction between live attenuated strains and lymphoid tissues, which is closely related to the immunogenicity of the live attenuated strain. Considering the bacterial colonization of C5000 (C500 *rpoS*^+^) in mice and the spleen pathological analysis of mice inoculated with mutant strains, it is unnecessary to reevaluate the colonization of mutant strains in mice. In in vivo experiments, all strains could induce a local mucosal immune response and systemic humoral immunity in mice at 9 weeks after the initial immunization. Mice were challenged with a lethal dose (10,000 × LD_50_) of wild-type *S*. Choleraesuis C3545 at 9 weeks after the initial immunization. The results showed that although C5001 (C500 *rpoS*^+^ Δ*crp10*) and C5002 (C500 *rpoS*^+^ Δ*fur9*) can protect against wild-type *S*. Choleraesuis in mice (Figure 4D), further enhancing the immune response is required.

It was reported that 50% of piglets showed a severely adverse reaction when the viability rate of the C500 attenuated live vaccine was less than 40%. The primary reason for the high side effects of C500 was the high endotoxin content released from the lysis of dead bacteria. Therefore, it is essential for a live bacterial vaccine to reduce the endotoxic activity when the vaccine is injected. Our previous research indicated that the deletion of *pagL*, *lpxR*, and codon-optimized *lpxE* substitution for *pagP* to engineer *Salmonella* synthesizing 4′-monophosphoryl-hexa-acylated lipid A decreases the interaction of lipid A and TLR4 and enhances the ability of *Salmonella* to adapt to the host environment, and the *Salmonella* vaccine with monophosphorylated lipid A showed low endotoxic activity while maintaining its good immunogenicity [31], which was also demonstrated by recent research conducted by Lee groups [51]. Therefore, the *lpxE* gene was introduced into constructing *S*. Choleraesuis vaccines to reduce endotoxin activity. The histopathological analysis in mice and rabbits demonstrated that overexpression of *lpxE* in the *S*. Choleraesuis vaccine resulted in decreased endotoxin activity, and it maintained its ability to induce an immune response in mice compared with the control groups C5002 (C500 *rpoS*^+^ Δ*fur9*) and SC30 (C500 *rpoS*^+^ Δ*crp10* Δ*pagL7* Δ*pagP81* Δ*lpxR9*) (Figure 5), indicating that the monophosphorylated lipid A in *S*. Choleraesuis is a suitable adjuvant that can reduce the toxicity of lipid A [31].

Our final purpose is to use attenuated live *S*. Choleraesuis as vectors to deliver heterologous antigens. One GDH protein antigen of *S*. *sui* and polysaccharide antigen of *E. coli* O9 O-antigen was chosen to evaluate its delivery ability. The results showed that SC3 D*asd* (C500 *rpoS*^+^ Δ*fur9* Δ*pagL7* Δ*pagP81*::P_lpp_
*lpxE* Δ*lpxR9* Δ*asd*) could effectively deliver the heterologous protein and induce a high level of antibody immune response in mice (Figure 7D). The results indicated that SC3 (C500 *rpoS*^+^ Δ*fur9* Δ*pagL7* Δ*pagP81*::P_lpp_
*lpxE* Δ*lpxR9* Δ*asd*) could be used as a carrier to develop multivalent vaccines. In this work, permanent inactivation of genes was included in our *S*. Choleraesuis construction. We are investigating and evaluating our delayed regulated attenuation system, which proved a good strategy for the live *S*. Typhimurium vaccine by Curtiss’ lab and us [39,52,53,54].

In conclusion, we constructed an ideal live attenuated *S*. Choleraesuis vaccine with low endotoxic activity. The key virulent gene *rpoS* in C500 was restored to an active state. Then, other essential virulent genes of *crp*, *fur*, *phoP*, and *aroA* were evaluated for their effects of deletion on safety and immunogenicity. Animal experiments showed that C5001 (C500 *rpoS*^+^ Δ*crp10*) and C5002 (C500 *rpoS*^+^ Δ*fur9*) showed an excellent ability to induce an immune response. To further decrease the endotoxic activity, the combination mutations, including Δ*pagL7* Δ*pagP81*::P_lpp_ *lpxE* Δ*lpxR9*, generating monophosphorylated lipid A, were introduced into the mutant strains. Animal experiments showed that SC3 (C500 *rpoS*^+^ Δ*fur9* Δ*pagL7* Δ*pagP81*::P_lpp_
*lpxE* Δ*lpxR9*) induced higher levels of IgG and secreted IgA antibodies and provided better immune protection, and it could be used as a vector for delivering heterologous antigens.

## Figures and Tables

**Figure 1 vaccines-12-00249-f001:**
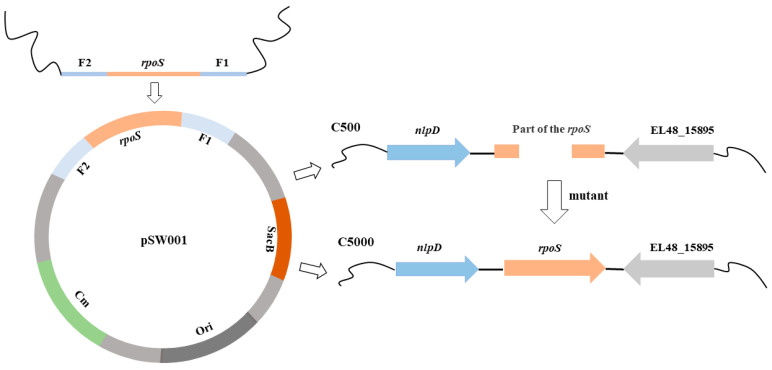
A schematic diagram of the restoring function of the *rpoS* gene of C500. The *rpoS* gene is partially missing in the C500 genome. The position of the *rpoS* gene in the C500 genome, and the position of the *rpoS* gene in the C500 genome after restoration.

**Figure 2 vaccines-12-00249-f002:**
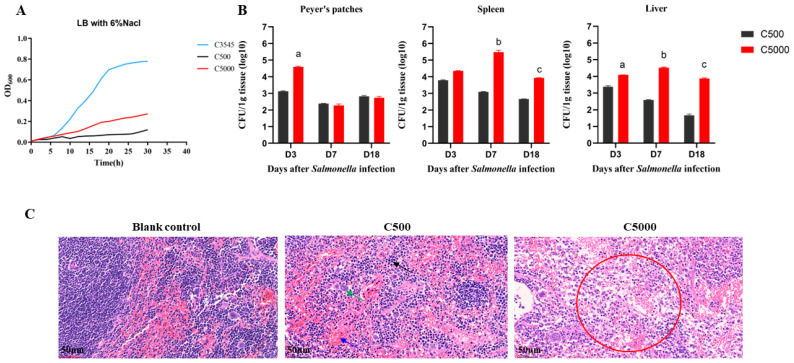
The biological characteristics and virulence analysis of the C5000 (C500 *rpoS^+^*). (**A**) Growth curves of *S*. Choleraesuis wild-type C3545, vaccine strain C500, and mutant strain C5000 (C500 *rpoS^+^*) in hypertonic LB broth (LB with 6% NaCl). (**B**) The bacterial load in mice liver, spleen, and PP at 3, 7, and 18 days after oral administration (*n* = 3). The total count of *Salmonella* colonies was calculated and presented as CFU per g of liver, spleen, or PP. (**C**) Histopathological assessment. *S*. Choleraesuis was orally administered to mice, and the effects on their organs were examined and contrasted with organ samples from animals given BSG as a placebo. Changes in the spleen’s histopathology were examined six days after administration. Blue arrows represent red blood cells, black arrows represent extramedullary hematopoietic cells, and green arrows represent macrophages. The red circle represents neutrophil infiltration. Images were captured with a magnification of 10× (scale bars 250 μm) and an inset magnification of 40× (scale bars 50 μm). Data are presented as the means ± SEM, and the superscript letters a, b, and c indicate *p* < 0.05 for comparisons with the C500 group at 3, 7, and 18 days, respectively.

**Figure 3 vaccines-12-00249-f003:**
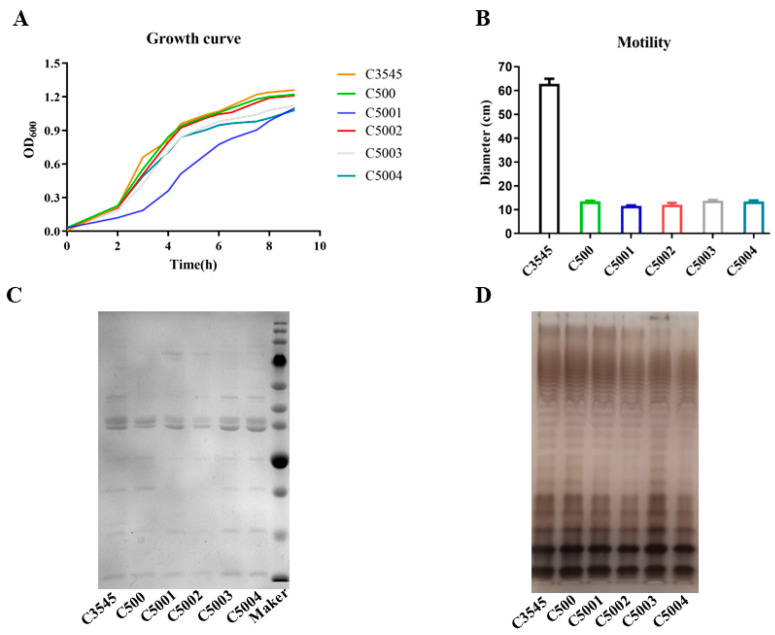
The growth phenotype of mutant strains in vitro. (**A**) Growth curves of wild-type C3545, vaccine strain C500, and mutant *S*. Choleraesuis strains C5001 (C500 *rpoS*^+^ D*crp10*), C5002 (C500 *rpoS*^+^ D*fur9*), C5003 (C500 *rpoS*^+^ D*phoP11*), and C5004 (C500 *rpoS*^+^ D*aroA12*) in LB broth. (**B**) A motility test of each attenuated strain. (**C**) The OMPs from the strains were isolated and analyzed on an SDS-PAGE gel staining by Coomassie Brilliant Blue. (**D**) The LPS profile of these strains was determined by silver staining.

**Figure 4 vaccines-12-00249-f004:**
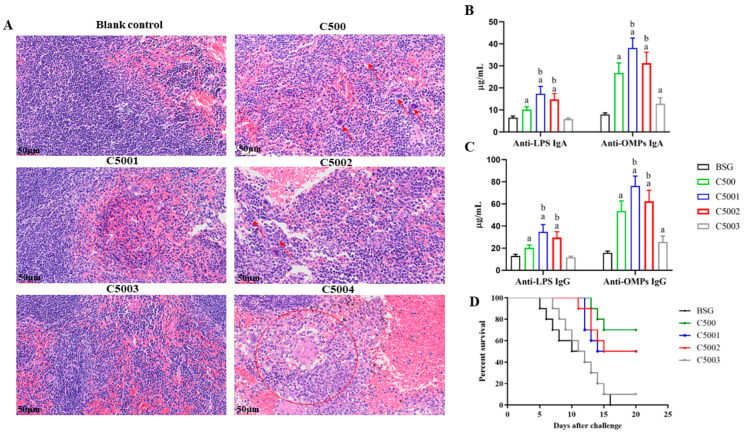
The histopathological analysis and immunogenicity assessment induced by C5001 (C500 *rpoS*^+^ Δ*crp10*), C5002 (C500 *rpoS*^+^ Δ*fur9*), C5003 (C500 *rpoS*^+^ Δ*phoP11*), and C5004 (C500 *rpoS*^+^ Δ*aroA12*). (**A**) The effects of recombinant attenuated *S*. Choleraesuis on organs were assessed by histopathological examination. The red arrows represent macrophages, and the red circles represent neutrophil infiltration. Images were captured with a magnification of 10× (scale bars 250 μm) and an inset magnification of 40× (scale bars 50 μm). (**B**,**C**) Recombinant attenuated *S*. Choleraesuis based on the C500 strain was investigated for immunogenicity by quantitative ELISA. The results displayed the precise levels of IgA and IgG antibodies, as measured by a standard curve, in mice orally inoculated with attenuated *S*. Choleraesuis at the scheduled weeks. The standard differences between the mice in each group were shown by the error bars. (**D**) Following the challenge, mortality was recorded for 25 days. Data are presented as the means ± SEM (*n* = 12), and the superscript letters a and b indicate *p* < 0.05 for comparisons with the BSG and C500 groups, respectively.

**Figure 5 vaccines-12-00249-f005:**
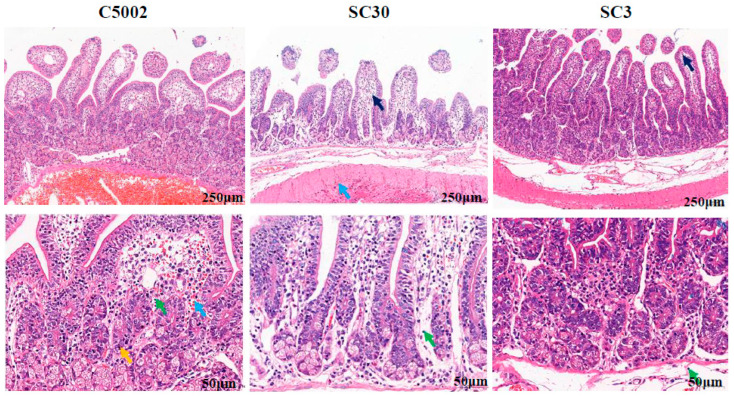
An ileal loop experiment on rabbits. Insertion of the *lpxE* gene on the *S*. Choleraesuis mutant’s chromosome reduces inflammation but stimulates fluid secretion in rabbit ileal loops. After injecting 10^9^ CFU of *S*. Choleraesuis strains into rabbit ileal loops for 8 h, the fluid secretion from the ileum was recorded, and the rabbit ileum was taken for H&E staining to observe histopathological abnormalities. The black arrows indicate tissue edema, the blue arrows indicate bleeding, the yellow arrows indicate neutrophils, and the green arrows indicate lymphocytes. Images were captured with a magnification of 10× (scale bars 250 μm) and an inset magnification of 40× (scale bars 50 μm).

**Figure 6 vaccines-12-00249-f006:**
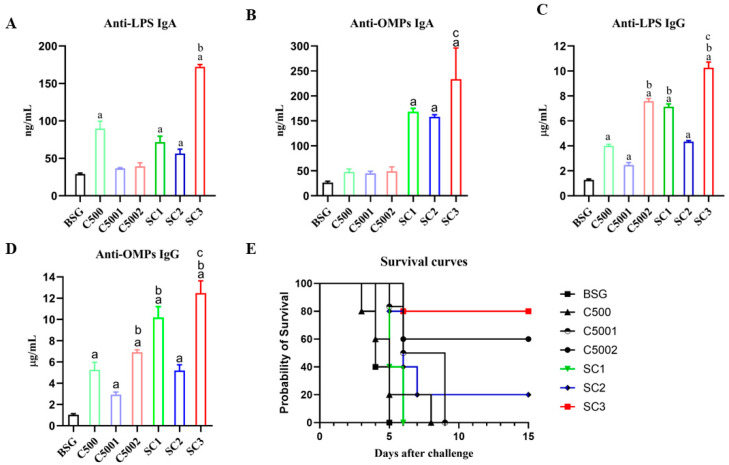
An immune response assay induced by recombinant attenuated *S*. Choleraesuis. Serum and vaginal secretions of mice were gathered at 8 weeks post-initial vaccination. Quantitative ELISA was applied to analyze specific IgA (**A**,**B**) and IgG (**C**,**D**) against *S*. Choleraesuis OMP and LPS. The results displayed the precise levels of antibodies, as measured by a standard curve, in mice orally inoculated with attenuated *S*. Choleraesuis at the scheduled weeks. The standard differences between the mice in each group were shown by the error bars. (**E**) Following the challenge, mortality was recorded for 25 days. Data are presented as the means ± SEM (*n* = 5), and the superscript letters a, b, and c indicate *p* < 0.05 for comparisons with the BSG, C500, and SC1 groups, respectively.

**Figure 7 vaccines-12-00249-f007:**
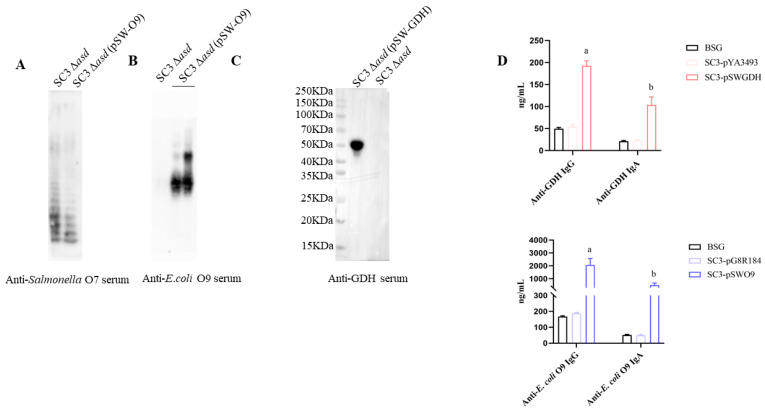
Evaluations of SC3 (C500 *rpoS*^+^ Δ*fur9* Δ*pagL7* Δ*pagP81*::P_lpp_
*lpxE* Δ*lpxR9*) as a vaccine carrier to deliver heterologous antigens. (**A**) The synthesis of *S*. Choleraesuis LPS in SC3 Δasd (pSW-O9). (**B**) The synthesis of *E. coli* O9 O-antigen in SC3 Δasd (pSW-O9). (**C**) The synthesis of *Streptococcus suis* GDH in SC3 Δasd (pSW-GDH). After strains SC3 Δasd (pSW-O9), SC3 Δasd (pSW-GDH), or SC3 Δasd reached an OD_600_ of 0.85 in LB broth, they were collected. For Western Blot analysis, the protein samples or LPS samples were separated on the SDS-PAGE, subsequently transferred onto nitrocellulose membranes, and analyzed with rabbit anti-*Salmonella* O7 serum, and rabbit anti-*E. coli* O9 serum, or rabbit anti-GDH serum. (**D**) Serum and vaginal secretions of mice were gathered at 8 weeks post-initial vaccination, and antibodies anti-GDH or anti-*E. coli* O9 LPS were detected by quantitative ELISA. The results displayed the precise levels of IgA and IgG antibodies, as measured by a standard curve, in mice orally inoculated with attenuated *S*. Choleraesuis at the scheduled weeks. The standard differences between the mice in each group were shown by the error bars. Data are presented as the means ± SEM (*n* = 5), and the superscript letters a and b indicate *p* < 0.05 for comparisons with the corresponding BSG group, respectively.

**Table 1 vaccines-12-00249-t001:** The bacterial strains and plasmids that were used in this study.

Strains or Plasmids	Description	Source
Plasmids		
pYA4278	*sacB mobRP4* R6K *ori* Cm^+^, pRE112	[36]
pYA4284	Δ*pagL*7 suicide plasmid	[31]
pYA4287	∆*lpxR*9 suicide plasmid	[31]
pYA4288	∆*pagP*8 suicide plasmid	[31]
pYA4295	∆*pagP* deletion, and P_lpp_ *lpxE* (codon-optimized) insertion	[31]
pSW001	*rpoS* gene insertion suicide plasmid	This study
pSW002	Δ*crp10* suicide plasmid	This study
pSW003	Δ*fur9* suicide plasmid	This study
pSW004	Δ*phoP11* suicide plasmid	This study
pSW005	Δ*aroA12* suicide plasmid	This study
pSS021	Δ*asd* suicide plasmid	[37]
pYA3493	Plasmid Asd^+^; pBRori β-lactamase signal sequence for periplasmic secretion	[38]
pG8R184	Asd^+^ vector, pSC101 ori, Kan^+^	[39]
pSW-GDH	Plasmid Asd^+^; pBRori β-lactamase signal sequence for periplasmic secretion, GDH of *S. sui*	This study
pSW-O9	Asd+ vector carrying genes involved in the biosynthesis of *E. coli* O9 O-antigen (*galF*-*gnd*), pSC101 ori	This study
*E. coli* strains		
χ7232	*endA1 hsdR17 (r_K_-,m_K_+) supE44 thi-1 recA1 gyrA relA1* Δ*(lacZYA-argF) U169* λ *pir deoR (*Φ*80dlac*Δ*(lacZ)M15)*	[40]
χ7213	*thi-1 thr-1 leuB6 glnV44 tonA21 lacY1 recA1 RP4-2-Tc*::μ λ *pir ∆asdA4 ∆zhf-2*::Tn*10*	[40]
Strains for immunization		
C500	C500 vaccine strain	China Veterinary Culture Collection Center
C5000	C500 *rpoS*^+^	This study
C5001	C500 *rpoS*^+^ Δ*crp10*	This study
C5002	C500 *rpoS*^+^ Δ*fur9*	This study
C5003	C500 *rpoS*^+^ Δ*phop11*	This study
C5004	C500 *rpoS*^+^ Δ*aroA12*	This study
SC1	C500 Δ*pagL7* Δ*pagP81*:: P_lpp_ *lpxE* Δ*lpxR9*	This study
SC2	C500 *rpoS*^+^ Δ*crp10* Δ*pagL7* Δ*pagP81*:: P_lpp_ *lpxE* Δ*lpxR9*	This study
SC30	C500 *rpoS*^+^ Δ*fur9* Δ*pag7L* Δ*pagP81* Δ*lpxR9*	This study
SC3	C500 *rpoS*^+^ Δ*fur9* Δ*pagL7* Δ*pagP81*:: P_lpp_ *lpxE* Δ*lpxR9*	This study
Strains for challenge		
C3545	Wild-type *Salmonella enterica* serotype Choleraesuis (*S.* Choleraesuis) strain, a clinical isolate from pig	S340 [41]

## Data Availability

The data presented in this study are available on request from the corresponding author. The data are not publicly available due to the ongoing expanded trial.

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
