# Peer review of "Construction and Evaluation of an Efficient Live Attenuated Salmonella Choleraesuis Vaccine and Its Ability as a Vaccine Carrier to Deliver Heterologous Antigens"

_vaccines, 2024, doi:10.3390/vaccines12030249_

Round 1

Reviewer 1 Report

Comments and Suggestions for Authors

Line 42….

 S. Choleraesuis, write the scientific name in italics in all the text.

Line 113…

The words in vitro and in vivo should be written in italics.

Line 120…

LB broth, LB should be written in the complete form in the first time

In figure 3C and 3D the sample are not indicated, there is only numbers from 1 to 6 with the same color.

Reviewer 2 Report

Comments and Suggestions for Authors

In this paper, the authors construct a series of mutant strains to obtain an efficient live attenuated Salmonella choleraesuis vaccine and finally, the SC3 vaccine (C500 rpoS+ Δfur9 ΔpagL7 pagP81:: Plpp lpxE ΔlpxR9) with good effect was constructed,  which is innovative. Subsequently, the authors again used SC3 as a vector to deliver heterologous protein antigens and polysaccharide antigens, and this section feels like that the author added it to increase the workload. In addition, there are the following places to recommend the author to modify:

1. In the 5-6,  Qing Liu 2, 3*, Qingke Kong 1 and 3*should be modified.

2. In the material and methods: (1) All reagents must be indicated with the manufacturer and batch number; (2) All the experimental methods were written too simply and needed to be detailed; (3) The author mentioned the lethal dose (1000 × LD50) in the results, so it is necessary to write clearly in the method how to determine the lethal dose (1000 × LD50).

3. In the results: (1) the name of the mutant strain is confusing, is it C5000 or C5000 (C500 rpoS+), same as C5001 (C500 rpoS+ Δcrp10) and C5002 (C500 rpoS+    Δfur9) , and it is suggested that the names of mutant strains should be unified in the whole text; (2) In this article, the results and discussion are written separately, and the results should not be compared and discussed with the results of others (line 283-289, line 382); (3) All legend descriptions should be placed below the figure; (4) Figure 7, A and b, missing Marker; (5) line 353, 109/CFU, the unit is wrong, may be ml/CFU.

Reviewer 3 Report

Comments and Suggestions for Authors

The authors constructed a new recombinant attenuated S. Choleraesuis vaccine with low endotoxin activity derived from vaccine strain C500. They evaluated its attributes both in vitro and in vivo by quantifiably comparing them to C500 to show attenuation and safety. They also assessed the ability of SC3 as a vector to deliver heterologous antigens for inducing immune responses in mice. These are important and interesting findings. However, there are concerns about the manuscript that should be addressed before being accepted for publication.

Major comments.

1.      Line 193: How many mice for each group were used to determine the virulence of mutants in mice? How was the LD50 calculated? What was the LD50 of the wild-type S Choleraesuis C3545?

2.      Line 220: What was the titer used to challenge the mice for the survival assay? How do you quantify the survival assay? Monitored for death daily? Please be specific.

3.      Lines 367, 428, 516: In lines 367 and 516, 1000XLD50 of the wild-type S Choleraesuis C3545 was used to challenge the vaccinated mice. However, in line 428, 10000XLD50 was used. Please clarify.

4.      Line 383: Why not use common biomarkers such as TNFα, IL1β, IL6, and IL8 to assess the induced inflammation?

5.      The work described in reference 54 has a similar goal to this paper. It would be much better to compare the data presented in both papers and discuss the new discovery of this work. For example, please explain why the authors in this paper use 1000xLD50 or 10000xLD50 to challenge the vaccinated mice. However, Dr. Shi’s lab only uses 20x to 40xLD50.

6.      Was Th1/Th2 immune responses evaluated?

Minor comments:

1.      Line 6: Qingke Kong1 and 3 should be changed to Qingke Kong1,3

2.      Table 1: C5000 should be included.

3.      Line 192: Please clarify the sentence “…..the bacterial loading was calculated based on the bacteria on the plates”.

4.      Lines 279-281: Figure legend should be positioned below the figure. This applies to all figures.

5.      Lines 308-319: C5000 should be included in Table 1 as described above.

6.      Figure 3: Please use the same font and font size in the same figure.

7.      Line 353: 109/CFU? Please correct this.

8.      Line 555: Curtiss's lab work was cited in references 43, 44, and 55, not in 52-54.  The work in Dr. Shi’s lab was cited in references 52-45/ Please correct it.

9.      Remove the duplicated number of references
